Assessing pollinators’ use of floral resource subsidies in agri-environment schemes: An illustration using Phacelia tanacetifolia and honeybees

Sprague Rowan 1 rowan.sprague@lincolnuni.ac.nz
http://orcid.org/0000-0002-0750-4864 Boyer Stéphane 1 2
Stevenson Georgia M. 3
Wratten Steve D. 1
1 Bio-Protection Research Centre, Lincoln University , Christchurch , New Zealand
2 Environmental and Animal Sciences, Unitec Institute of Technology , Auckland , New Zealand
3 Department of Ecology, Lincoln University , Christchurch , New Zealand
Benelli Giovanni
Electronic publication date: 2016 Nov 15
Publication date: 2016
Volume: 4
Electronic Location ID: e2677
Received 2016 Jun 27; Accepted 2016 Oct 12
Copyright: © 2016 Sprague et al.
Copyright year: 2016
Copyright holder: Sprague et al.
License: This is an open access article distributed under the terms of the Creative Commons Attribution License, which permits unrestricted use, distribution, reproduction and adaptation in any medium and for any purpose provided that it is properly attributed. For attribution, the original author(s), title, publication source (PeerJ) and either DOI or URL of the article must be cited.
License URL: https://creativecommons.org/licenses/by/4.0/

Keywords: Apis mellifera, Honeybee foraging behaviour, Agroecosystems, Pollen preference, Floral enhancements, Pollinator health strategies

Funding: Fulbright New Zealand and Fulbright US Bio-Protection Research Centre Fulbright New Zealand and Fulbright US funded Rowan’s US Graduate Award, enabling her to complete this research in NZ. Bio-Protection Research Centre funded Georgia’s summer scholarship. The funders had no role in study design, data collection and analysis, decision to publish, or preparation of the manuscript.

==============================
Background

Honeybees (Apis mellifera L.) are frequently used in agriculture for pollination services because of their abundance, generalist floral preferences, ease of management and hive transport. However, their populations are declining in many countries. Agri-Environment Schemes (AES) are being implemented in agricultural systems to combat the decline in populations of pollinators and other insects. Despite AES being increasingly embedded in policy and budgets, scientific assessments of many of these schemes still are lacking, and only a few studies have examined the extent to which insect pollinators use the floral enhancements that are part of AES and on which floral components they feed (i.e., pollen and/or nectar).

Methods

In the present work, we used a combination of observations on honeybee foraging for nectar/pollen from the Californian annual plant Phacelia tanacetifolia in the field, collection of pollen pellets from hives, and pollen identification, to assess the value of adding phacelia to an agro-ecosystem to benefit honeybees.

Results

It was found that phacelia pollen was almost never taken by honeybees. The work here demonstrates that honeybees may not use the floral enhancements added to a landscape as expected and points to the need for more careful assessments of what resources are used by honeybees in AES and understanding the role, if any, which AES play in enhancing pollinator fitness.

Discussion

We recommend using the methodology in this paper to explore the efficacy of AES before particular flowering species are adopted more widely to give a more complete illustration of the actual efficacy of AES.

Introduction

As many as 70% of crop species worldwide benefit directly or indirectly from pollination by animals (Klein et al., 2007), with insects contributing the most to this ecosystem service (ES). Of these pollinators, honeybees (Apis mellifera L.) are used most frequently in agriculture for pollination services because of their abundance, generalist floral preferences, ease of management and transport of hives, and revenue-generating by-products (Tautz, 2008; Aizen & Harder, 2009; Potts et al., 2010a). Although the significance of unmanaged insects in crop pollination has been increasingly recognised (Winfree et al., 2008; Rader et al., 2009; Woodcock et al., 2013), reliance on honeybees has increased in recent decades in response to rising pollination needs and overall population declines of pollinators (Aizen et al., 2008; Breeze et al., 2011).

While demand for this ES is high, honeybee populations are declining in many countries, including the United States, Canada, the UK and Germany (Potts et al., 2010a; Potts et al., 2010b; van der Zee et al., 2012; vanEngelsdorp et al., 2012). No single cause of this decline has been identified; rather, several factors are involved, notably varroa mites (Varroa destructor) (Sammataro, Gerson & Needham, 2000; Shen et al., 2005), pathogens such as American foulbrood (caused by the bacterium Paenibacillus spp.) (Genersch, 2010), fungal parasites such as Nosema (Potts et al., 2010a; Pettis et al., 2013), loss of biodiversity and associated floral resources in agricultural systems (Kremen et al., 2007; Klein et al., 2007; Potts et al., 2010a), and the use of neonicotinoid pesticides (vanEngelsdorp & Meixner, 2010; Potts et al., 2010a; Goulson, 2013).

Agri-environment schemes (AES) have been developed to mitigate some of these declines in ES, and this paper provides an illustration of how the value of such schemes for pollinators can be better assessed.

What are agri-environment schemes, and how effective are they?

In response to pollinator population declines, the European Commission has recommended programmes aimed at improving pollinator fitness and efficacy as part of the wider fiscal policy of AES. Originating from the 1980s to protect biodiversity and important cultural areas in England, these schemes attempt to inform landowners and policy makers on methods to manage land sustainably and enhance ES (Natural England, 2012). Included in this policy are methods that claim to benefit pollinators such as bees and butterflies (Natural England, 2013a; Natural England, 2013b). The methods recommended to improve pollinator fitness and efficacy do not usually involve crops which need to be pollinated by insects, but rather are usually a mix of wildflowers selected because of their high nectar and/or pollen quality planted in the field margins beside crops. AES have evolved over time to incorporate more tractable ways for farmers to become involved and to include more management recommendations to enhance a range of multiple ES; examples include ‘beetle banks’ or habitats for beetles (Thomas, Wratten & Sotherton, 1992) and buffer strips of wildflowers between croplands (Department for Environment, Food Rural Affairs & UK Government, 2014). By 2009, about 66% of England’s agricultural land was managed as part of AES agreements (Natural England, 2009). Other countries are also attempting to address the issue of pollinator decline. Recently the US Federal Government also proposed the Pollinator Health Strategy 2015, which seeks to improve the health of honeybees and native pollinators, and restore land to be used as pollinator habitat. Differing from the AES Environmental Stewardship Handbooks which guide farmers on how to implement AES, this report acknowledged the elements in its strategy which need to be researched before the government can recommend more effective policies (Vilsack & McCarthy, 2015).

Despite AES being increasingly embedded in policy and budgets, scientific assessments of many of these schemes still are lacking, such as quantifying the type, extent and quality of ES actually delivered (Kleijn & Sutherland, 2003; Kleijn et al., 2006; Whittingham, 2007). Also, few studies have examined the extent to which insect pollinators use the floral enhancements and whether the added resources serve as shelter, nest sites or pollen/nectar resources. In a study by Holland et al. (2015), researchers found that farm management, such as establishing florally-enhanced grass areas, and the amount of uncropped habitats can positively affect the species richness and abundance of insect populations. An example of an AES approach which specifically targets pollinators is that by Pywell et al. (2006), who found that bumblebees (Bombus spp.) benefited from wildflower mixes and mixes of flowers with known high-quality pollen and nectar through increased abundance and species richness. However, this study did not examine how the Bombus spp. used the added floral resources and whether the bumblebees preferred the added flowers. This raises a key issue with studies of insect pollinators through floral enhancement schemes. These schemes often lack comparisons of how these insects use the added floral resources compared to the other surrounding flowers. For example, there need to be more studies assessing the extent to which pollinators ‘prefer’ particular added flowers, their relative use of pollen and nectar on each plant species visited, diurnal and seasonal use of the resource, the value of the added flowers for other beneficial arthropods, and any associated ecosystem disservices (e.g., weediness potential) if these potential AES are to be adopted (Zhang et al., 2007; Wielgoss et al., 2014). Our study aims to addresses a key part of this issue, that of examining how and to what extent honeybees use the Californian annual plant Phacelia tanacetifolia (Bentham: Borginaceae; tansy leaf).

Farmers and policy makers need to have a clear and straightforward way to assess the values of AES implemented on farms. In the present work, we used a combination of observations on honeybee foraging for nectar/pollen from Phacelia tanacetifolia and collection of pollen pellets from nearby hives to assess the value of adding phacelia to an agro-ecosystem to benefit honeybees.

Honeybees as the study organisms

Honeybees were used here not only because of their agricultural importance, but also because of their distinct foraging behaviours. Individuals collect pollen by gathering the grains from the anthers of flowers and they use nectar to keep the grains together. They carry the pollen in pollen baskets (corbiculae) on their hind legs, forming pellets of pollen grains (Tautz, 2008). These bees demonstrate floral constancy, meaning that individual foraging bees will visit only one species of flower on any one day, sometimes even over several days (Free, 1963). As a result of this constancy, there is a 95–99% likelihood that a pellet will comprise pollen from only one species (Tautz, 2008).

There are a few studies that explore the use of AES by honeybees and the potential ways in which they benefit from them. Couvillon, Schürch & Ratnieks (2014) examined honeybee waggle dance patterns to determine where they forage in landscapes containing different types of AES, as well as areas without any such stewardship measures. They found that these insects showed a significant foraging ‘preference’ for Higher Level Stewardship sites, or sites that met more complex requirements to address local needs and provide more than one ES (Couvillon, Schürch & Ratnieks, 2014). However, this study did not explore whether or not this ‘preference’ was correlated with the type and quantity of floral resources available. Balfour et al. (2015) also examined honeybee waggle dance patterns and the surrounding available flowers for pollinators to determine where honeybees were foraging and which habitat and flower types in which they were most abundant. They found that honeybees were mostly found in field margins and hedgerows and that they foraged mainly for agricultural weeds. This study did not examine specifically whether honeybees preferred certain flowers over others and it did not take place where flowers had been added to enhance honeybee fitness or efficacy. Carvell et al. (2007) found that bumblebee species abundance and diversity increased in response to field margins with legumes providing pollen and nectar, but they concluded that a more diverse mix of flowers should be planted to offer a range of blooming durations and flower phenology. Few studies have examined the foraging preference of honeybees, or lack thereof, for particular plant species providing floral resources in an agricultural context, and no studies have examined honeybee preference for floral resources in AES guidelines to test and better inform AES field and crop margin design. We did not study wild bees (native or bumblebees) in this study, as assessments of their preferences and colony fitness would be difficult due to their different life cycles and access to their nests compared to managed honeybee colonies.

Here, we used phacelia as a potential supplementary floral resource because this species is commonly included in florally-enhanced field margins AES (Carreck & Williams, 2002; Decourtye, Mader & Desneux, 2010). It is a high-quality honey plant (Crane, Walker & Day, 1984) and a wide range of insect species forage on it (Carreck & Williams, 1997). Its pollen is said to have a high protein content (Trees for Bees, 2014) and its nectar and pollen improve the predation rate of insect biological control agents, including hoverflies (Diptera: Syrphidae) (Hickman & Wratten, 1996; Laubertie, Wratten & Hemptinne, 2012). For example, Laubertie, Wratten & Hemptinne (2012) compared how six different flowers (phacelia, buckwheat, coriander, alyssum, mustard, and marigold) commonly used as floral enhancements improved the fitness of hoverflies, and they found that phacelia overall improved fitness the most. For these reasons, we chose to use phacelia as our test flower species.

Materials and Methods

Field site

The field site was located on the Canterbury Plains, New Zealand within an agricultural landscape (latitude: −43.63788760, longitude: 172.53225660). A single strip of P. tanacetifolia cv. Balo 8 m long and 1.5 m wide was sown every two weeks to ensure a continuous flowering period of at least six weeks. The honeybee hives used for this experiment were located about 25 m from the sown flowers. Two healthy hives with queens of similar age of 1–2 years old (determined by the beekeeper) were chosen, as the age of queens affects the pollen demand of the colony and thus the extent of pollen foraging (Tautz, 2008). While the site was modelled after AES guidelines for the UK, recommendations of enhancing agricultural landscapes with wildflowers or known insect-beneficial flowers are similar throughout many countries. Therefore the results from this study should be generalizable to other countries.

Pollen collection

Pollen traps (Dimou & Thrasyvoulou, 2007) were installed on both hives. Pollen pellets were collected every day on which no rain had fallen and which had a maximum air temperature at or above 14 °C (a total of twelve days between 7 November and 9 December 2014). Honeybees do not leave the hive to forage when the outside air temperature is below 11 °C or it is raining (Dimou, Thrasyvoulou & Tsirakoglou, 2006). The main entrance of the pollen traps was closed at 11:00 h and opened again at 13:00 h to collect all the pollen which the foraging bees brought back to the hive during this 2-h period. This period was selected because honeybee foraging activity is high during the middle of the day (García-García, Ortiz & Dapena, 2004) and because this period was likely to include those plant species for which the bees foraged in the morning and the afternoon.

The pollen pellets from each of the hives were collected at the end of the 2-h period and placed in separate 25 mL containers and stored at −20 °C to eliminate fungal growth. The pollen pellets from one of the hives during one of the 2-h periods was considered to be one sample.

Species identification from pollen samples

The pellets in each sample were weighed together and counted. The purple ones were separated from the others using forceps and placed in individual tubes, as the purple colour is indicative of P. tanacetifolia pollen and no other known flowers in the area were thought to produce purple pollen. To confirm that the purple pellets collected really did comprise phacelia pollen, DNA barcoding with ITS primers was used to verify the species identity of the P. tanacetifolia pollen pellets.

Observations of honeybees foraging on P. tanacetifolia

The extent of the use of P. tanacetifolia by honeybees was examined through the pollen brought back to the hive and through the worker bees’ behaviour in the field.

The numbers of honeybees foraging for nectar and pollen, respectively, were observed visually and counted in a 10 m2 area over 5-min periods at 10:00, 12:00, and 15:00 for ten days between mid-December to end of January depending on weather conditions. Bumblebees were also visually counted to see whether other pollinators were also foraging on the phacelia. Twenty flowers of P. tanacetifolia were chosen randomly and the quantity of pollen on the anthers was scored; the scoring scale is shown in Table 1. The ages of the P. tanacetifolia flowers were also scored using the four stages defined by Williams (1997); this scoring scale is shown in Table 2.

Table 1 Scores for amount of pollen on P. tanacetifolia flowers.

In brackets: percentage of the anther covered with pollen.

Amount of pollen	Score	
No visible pollen (0%)	0	
Small amount of visible pollen (25%)	1	
Some visible pollen (50%)	2	
Large amount of visible pollen (75–100%)	3	

Table 2 Score for maturity of P. tanacetifolia flowers (from Williams, 1997).

Maturity of flowers	Score	
Just-opened flower (Stage 1: curled filaments and style)	1	
Mid-age flower (Stage 2: filaments uncurled and petals at about 60°)	2	
Mid-age-old flower (Stage 3: petals at about 20–60°, styles longer than filaments)	3	
Older flower (Stage 4: petals closing, some anthers may have fallen off the filaments)	4	

Since the area of P. tanacetifolia flowers blooming changed throughout the experiment because of the sequential drilling, honeybee counts were divided by the area of flowers present on each date.

Data analysis

For the pollen pellet collection experiment, we analysed the count data to determine whether the total number of phacelia pollen pellets collected was significant compared to the total overall number of pellets. For our field observation experiments, to see if the number of honeybees foraging for nectar differed from the number of honeybees foraging for pollen, we used a paired two-tailed t-test. To test whether the number of honeybees foraging for nectar and for pollen varied with the time of day, the maturity of the flowers, and the amount of pollen on the flowers, a mixed effects linear model was used to account for the day on which the observations were taken. The data were bootstrapped to determine the 95% confidence intervals around the data. Likelihood Ratio Tests were used to determine which of these factors (if any) was significant for pollen and nectar foraging bees. We also used a mixed effects linear model to test whether the total number of bees foraging for both nectar and pollen was affected by the time of day, maturity of the flowers, and the amount of pollen on the flowers. We used a Likelihood Ratio Test again to determine whether these factors were significant overall. The R software program was used to explore and analyse the data (R Development Core Team, 2014).

Results

Pollen collection experiment

Only one P. tanacetifolia pollen pellet was found in a total of 23,431 pellets collected. The results of the DNA barcoding analysis confirmed that the purple pellet was from P. tanacetifolia. A preliminary DNA barcoding analysis of other pollen pellets showed that the honeybees also collected pollen from clover (Trifolium spp.), dandelion (Taraxacum spp.) and brassicas (Brassica spp.). No statistical tests for significance were run on these data as the number of P. tanacetifolia pollen pellets was negligible.

Honeybees foraging on P. tanacetifolia in the field

The virtual absence of P. tanacetifolia pollen collected by honeybees is supported by the observations of their foraging behaviour in the field. Using a two-tailed t-test, the number of honeybees foraging for pollen was significantly lower than the number of those foraging for nectar (p value < 0.001). Figure 1 shows the mean number of honeybees foraging for pollen and those foraging for nectar at different times of day. Comparing these results with those from the pollen collection experiment showed that while honeybees may virtually ignore the pollen of P. tanacetifolia, they readily use the flowers for nectar.

Figure 1 Mean numbers of honeybees foraging for nectar and for pollen at 10:00, 12:00, and 15:00 (mean ± standard error; n = 12).

The boxplots in Fig. 2 illustrate the relationships between the number of foraging honeybees, pollen amount, and flower maturity. As there were minimal numbers of honeybees foraging for pollen, Fig. 2A does not show any significant differences between the pollen amounts on the flowers. The median number of honeybees foraging for pollen and nectar did not differ between the four stages of flower maturity (Fig. 2B).

Figure 2 Boxplots showing relationships between foraging honeybees and P. tanacetifolia.

(A) the relationship between the number of honeybees foraging for pollen and the amount of pollen found on the P. tanacetifolia anthers (n = 10) and (B) the relationship between the number of honeybees foraging for both pollen and nectar, and flower maturity (n = 10). No significance was found between the groups for A and B. On the boxplots, the box regions delineate the 25-75% quantiles of the data and the bold black line represents the median of the data. The “whiskers” of the boxplot represent the 95% confidence interval of the median, and the open circles represent the outliers in the data.

The results of Likelihood Ratio Test from the linear mixed effects model of number of pollen-foraging bees modelled by time of day showed that time was not significant (test = ANOVA, df = 4, 6, χ2 = 2.132, p value = 0.344) although there was a decrease in the number of pollen-foraging bees at 12:00 and 15:00. Neither pollen amount nor flower maturity had a significant effect on the number of pollen foragers (test = ANOVA, df = 5,10, χ2 = 7.749, p value = 0.171; test = ANOVA, df = 6, 10, χ2 = 5.648, p value = 0.227 for pollen amount and flower maturity, respectively).

For the linear mixed effects models of nectar-foraging bees, the results of Likelihood Ratio Test for time of day showed that time was not significant (test = ANOVA, df = 5, 10, χ2 = 4.984, p value = 0.418) although there was a decrease in the number of nectar-foraging bees at 12:00. Flower maturity did not have a significant effect on the number of nectar foragers (test = ANOVA, df = 6, 10, χ2 = 4.988, p value = 0.289) and there was not a noticeable trend in the data either. For pollen amount on the flowers, there was a trend with increasing numbers of nectar-foraging bees with higher amounts of pollen; however this trend was not significant (test = ANOVA, df = 5, 10, χ2 = 1.913, p value = 0.861).

For the linear mixed effects models of the total number of foraging bees (for both pollen and nectar), the results were similar to that of the nectar-foraging models. The results of Likelihood Ratio Test for time of day showed that time was not significant (test = ANOVA, df = 5, 10, χ2 = 7.141, p value = 0.210) although there was a decrease in the number of foraging bees at 12:00 and 15:00. Flower maturity did not have a significant effect on the number of total foragers (test = ANOVA, df = 6, 10, χ2 = 7.004, p value = 0.136). For pollen amount on the flowers, there was again a trend with increasing numbers of foraging bees with higher amounts of pollen; however, this trend was not significant (test = ANOVA, df = 5, 10, χ2 = 2.862, p value = 0.721).

Discussion

AES are being implemented in agricultural systems to combat the decline in populations of pollinators and other insects, such as butterflies, as well as birds and mammals (Fahrig, 2003; Biesmeijer et al., 2006; Park, 2015; Frenzel, Everaars & Schweiger, 2016). The work here demonstrates that for honeybees, pollen identification at the hive can aid assessments of the extent to which pollen is being used by pollinators. Combining this method with detailed observations in the field can give a more complete illustration of the actual efficacy of a particular floral resource.

The results here, showing honeybees gather mainly nectar from P. tanacetifolia, are supported by the study by Williams & Christian (1991) which found that most visits by honeybees to this plant were for nectar rather than pollen (nectar 78/22% pollen), although this study did not study P. tanacetifolia in an AES context. However, P. tanacetifolia is used in crop margins in AES because its pollen has a high protein content and that this resource, as well as its nectar, can improve the fitness and longevity of insects (Hickman & Wratten, 1996; Laubertie, Wratten & Hemptinne, 2012). Because this current work has shown that honeybees used nectar almost exclusively, the plant’s benefits to bees may be less than those expected when AES protocols are designed. Although the nectar that the honeybees gathered from the P. tanacetifolia flowers may have benefited the honeybee colonies, the bees were not receiving the expected benefits of the high-quality P. tanacetifolia pollen.

This study also examined honeybee foraging behaviour on the P. tanacetifolia in relation to pollen amount and flower maturity. There was no significant relationship between the number of honeybees foraging on the phacelia and the amount of pollen found on the flowers, nor flower maturity for pollen, nectar, and total number of foraging bees. This may be because the number of bees foraging on the phacelia was so low overall that no significant conclusions could be drawn from this aspect. While there was no significant difference between the number of bees foraging and time of day, there did appear to be a trend for more foragers at 10:00 than at 12:00 and 15:00. One explanation for this apparent decrease is that pollen was depleted in the morning and not fully replenished later in the day.

Another explanation for decreasing foraging activity later in the day (and for low rates of foraging on P. tanacetifolia in general, with a maximum foraging density of only 5.4 honeybees per m2) is that this plant was a limited resource. Because honeybees as a colony tend to prefer to forage on the same species of plant when it’s in high densities (Tautz, 2008), the P. tanacetifolia may not have been a large enough resource to provide a large or consistent enough reward to the honeybee colonies. To test whether this plant can be a limited resource or is of inherently low preference by honeybees, it should be planted as a mass flowering crop, as recommended by Westphal, Steffan-Dewenter & Tscharntke (2003), to provide enough resource to be a sufficiently abundant resource for honeybees. In that case, similar methods to those carried out here can be used. A recent study by Henry et al. (2012) planted a total of 1.2 km2 of P. tanacetifolia strips in a 5 km radius around honeybee hives, from which pollen was collected. That study found as much as 32% of the pollen that was collected from the hives was from the P. tanacetifolia strips, which supports the mass flowering idea.

It follows that AES may be of limited efficacy for honeybees because of floral area; most that have been implemented are not appropriately monitored and evaluated to test for effects of floral area planted on the response of insect pollinators (Kleijn & Sutherland, 2003; Kleijn et al., 2006; Whittingham, 2007). For example, both the United States Federal Government’s Pollinator Health Strategy report (Vilsack & McCarthy, 2015) and AES Environmental Stewardship Handbooks (Natural England, 2013a; Natural England, 2013b) lack practical and scientific ways to record and monitor the results of the landscape ‘enhancements’ made. Therefore, we recommend that the approach used in this study be used by researchers and farmers’ advisors to assess AES in more detail to determine how pollinators use them and how much they actually benefit from them. We suggest that the following procedures need to be included in AES assessments: Use standard and repeatable field observations of how pollinators use the added floral enhancement in the field.

Identify the pollen that foraging bees bring back to the hive to determine the main plant species on which the bees forage and the extent to which pollen comes from the added floral resources.

Components 1 and 2 above were addressed in this study and were crucial for identifying how the honeybees were using the phacelia. By including observations of honeybees in the field and quantitative data from pollen brought back to the hive, scientists and stakeholders can determine the extent to which the honeybees were using the floral enhancements mainly for pollen or nectar, whether there were any temporal trends, and for which plant species the insects were mainly foraging.

An assessment of ‘fitness’ should also be considered as a component of long-term AES assessments, especially because these schemes have been proposed to enhance pollinator fitness. Such fitness improvements, or proxies for them, are most likely to be expressed in female worker bee numbers, the quantity of honey stored, survival rates in the colony through winter, and the extent to which the colony produces swarms in the spring. Assessments focussing on unmanaged or wild bees such as bumblebees, should make us of fitness indicators suitable for these particular species.

Conclusions

The experiments here, which examined honeybees’ use of P. tanacetifolia, found that the insects hardly use the flowers as a pollen resource although nectar was readily taken. This work lays the foundations for future research aiming to evaluate the benefits of particular AES and other schemes which aim to enhance on-farm functional biodiversity. We provide a framework for the experimental design and analysis of the actual benefits to pollinators and other beneficial insects. Increased understanding of how pollinators use these potentially high-quality floral resources added to landscapes will lead to better-designed pollinator enhancement schemes and potentially higher fitness and populations of pollinators.

Supplemental Information

Supplemental Information 1 Raw data.

Data was collected at our field site at 10:00, 12:00, and 15:00 and the plot size of flowering phacelia was recorded in square meters. The following environmental variables were measured: cloud cover, temperature, wind speed, and soil moisture. Estimates of Flower Maturity and Pollen Amount were made using Tables 1 and 2 found in our manuscript. Bees.Nectar refers to the number of honeybees which were foraging for nectar counted during a five minute interval in the plot. Bees.Pollen refers to the number of honeybees foraging for pollen during the same five minute interval in the plot. These counts were totalled to get the column Bees.Total.

Click here for additional data file.

The authors would like to thank the Bio-Protection Research Centre for its support through excellent staff and equipment. We would also like to thank the Field Service Centre at Lincoln University, in particular Alan Marshall, Dave Jack, and Dan Dash, for helping cultivate the phacelia at our site. Finally, we would especially like to thank Dan Arthur, the beekeeper and landowner of the experimental site who provided beehives for the experiments and gave us valuable beekeeping and honeybee advice.

Additional Information and Declarations

Competing Interests

Author Contributions

Field Study Permissions

Data Deposition

The authors declare that they have no competing interests.

Rowan Sprague conceived and designed the experiments, performed the experiments, analyzed the data, wrote the paper, prepared figures and/or tables, reviewed drafts of the paper.

Stéphane Boyer conceived and designed the experiments, reviewed drafts of the paper.

Georgia M. Stevenson conceived and designed the experiments, performed the experiments, reviewed drafts of the paper.

Steve D. Wratten conceived and designed the experiments, reviewed drafts of the paper.

The following information was supplied relating to field study approvals (i.e., approving body and any reference numbers):

Dan Arthur, the beekeeper and landowner of the experimental site, provided beehives for the experiments.

The following information was supplied regarding data availability:

The raw data has been supplied as Supplemental Dataset Files.

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
