# Peer review of "Assessing pollinators’ use of floral resource subsidies in agri-environment schemes: An illustration using Phacelia tanacetifolia and honeybees"

_PeerJ, doi:10.7717/peerj.2677_

## Round 0.1 · original submission · Major Revisions

Dear Authors,

I have received different recommendations from three Reviewers, ranging from Minor Revisions to Major Revisions and Reject. In particular Reviewer 2 raised severe criticisms on the technical quality of the ms, while Reviewer 1 asked for more details about the data analysis. Therefore, I encourage you to improve the manuscript following Reviewers' comments. Then, I will take a final editorial decision on it.

Reviewer 1 ·

Basic reporting

Overall, the text is clear and unambiguous. However the Introduction is long and does not clearly state the aim of the study. The authors’ goal was to demonstrate that before flowers are planted in Agri-Environmental Schemes (AES), they should be tested for their utility and benefit to bees. The authors point out a gap in our understanding, that we currently have limited evidence that links AES with the outcomes they are meant to promote. But the purpose of the AES should be stated: They provide forage (flowering resources) for honey bees (and other bees) so that the bees are well-nourished and able to pollinate crops, presumably later in the season. The AES’s are not crops that require pollination, but are wildflowers that benefit by being pollinated by bees.
They give several examples of studies, which are well cited, and raise two "key issues" based one study (Pywell et al. 2006). Their first key issue that effects on biodiversity may not translate into effects on pollination services is a valid point. However, the author's experiment was not designed to assess pollination services. Also, if they are interested in determining a framework for assessing the effectiveness of AES's that aim to enhance pollination services, then the second key issue is not relevant.
They gave a helpful explanation of why honey bees and Phacelia lend themselves to testing their goal, but the section about studies on AES and honey bee foraging preferences appears to be missing some information. While the authors did a nice job of describing the Couvillon et al. (2014) study, they assert that the study did not examine the type and quality of floral resources without mentioning the follow-up study, Balfour et al. 2015 (Agriculture, Ecosystems and Environment 213: 265–271) that did examine floral resources and honey bee flower visits. In addition, the statement that "No studies have examined the foraging preference of honey bees, or lack thereof, for particular plant species providing floral resources in mixtures in AES" requires a little clarification. Are you only considering official AES in the UK or all landscape enhancements for pollinators? Is relative visitation enough to assess preferences or is it necessary to determine the percentage of the total colony diet made up by a particular species of plant? Otherwise, the statement might seem to dismiss some studies of bee visits to flowers in landscape enhancements in the US (ex. Morandin and Kremen 2012).

In the Abstract and other places in the manuscript, (e.g, Line 118) they refer to “pollen basket” which is the common name for corbicula, where bees carry pollen on hind leg. The authors collected pollen loads from the corbiculae of the bees (they did not collect pollen baskets).
The figures have clear text and the legends are easy to read. The captions provide all necessary information except the caption for Figure 2 needs sample sizes
Minor corrections:
• Line 85- between croplands
• Line 102- comma instead of a period
• Line 244- limiting resource?
• Line 249- a sufficient resource
• Line 256- "for this aspect" -> "to test for effects of area"
• Line 285- "proposed to pollinators' ecological traits" -> "proposed to enhance pollination fitness" or "proposed to enhance pollination services"

Experimental design

This study was original, primary research within the scope of PeerJ.
In this study, they tested one flower, Phacelia tanacetifolia -- commonly planted in AES because it provides pollen (protein/ lipids) for insects -- to determine if honey bees collect pollen from it. They found, for the plot size they planted, honey bee visited this species more for nectar than for pollen. As honey bees require lots of nectar to convert to honey, it seems Phacelia does benefit honey bee productivity and does benefit them, a point not made in this manuscript.

One confusion in experimental design stemmed from the authors' desire to talk about two criteria for assessing AES that their experiment did not address (Discussion, lines 265-277 and Introduction, lines 104-107). They would have a tighter argument if they chose to focus only on their first two criteria.
The experimental design seems reasonable for this question, although it might have been better to test different plot sizes of Phacelia to address some of the concerns about floral area they bring up in Discussion (e.g., line 255).
Also, it is great that Phacelia pollen is such a distinctive color, making it good choice of flower species to test. If the authors have honey samples from these colonies, it would be good to spin the pollen in the honey down and look at it under the microscope. P. tanacetifolia honey has a high concentration of pollen grains (Bryant and Jones 2001) so they could look at its contribution to nectar stores if they corrected using the published Phacelia pollen coefficient.
The methods seemed thorough except that the statistical software was missing. I would recommend adding that and putting the code in the supplementary material. The tables defining the scoring systems were very helpful.

Validity of the findings

I could not open the raw data file. Without the raw data and more knowledge of the statistical tests used, this is hard to judge, but the sample sizes were given (two hives, 23,431 pollen pellets, and 36 observations of pollen/nectar foragers). The striking pollen trap result is very clear regardless of the tests used.
The authors bring in useful citations to discuss the potential effect of the total number of flowers on honey bee recruitment and foraging preferences. They made a good argument, especially when comparing their results to the results of Henry et al. (2012). But as stated above, it is important to note that nectar sources are very important, even if they don't provide pollen. The discussion of assessing colony fitness and non-target species biodiversity strays from the results, and I think it would be a stronger article if they focused on their first two procedures. I would at least suggest that they make a distinction between assessments of honey bee and wild bee fitness. For example, beekeepers work hard to suppress swarming behavior. Colony reproduction would be more useful in assessing effects on bumble bees.

Reviewer 2 ·

Basic reporting

The manuscript presents data from field experiments among which authors used a combination of observations on honeybee foraging for nectar/pollen from the plant Phacelia tanacetifolia in the field and collection of pollen baskets from hives, to assess the value of adding phacelia to an agro-ecosystem to benefit honeybees. They showed that honeybees may not use the floral enhancements added to a landscape as expected and points to the need for more agri-environment schemes assessments and understanding the role that such schemes play in enhancing pollinator fitness.

Experimental design

Materials and methods: in general they are lacking in technical quality and appear too vague in many parts. The main criticism I noted is about the lacking of necessary details on the statistical approach you used, that should be clearly presented in a dedicated paragraph. Moreover, I feel that one year of observation is too scanty to provide adequate information on the topic.
Specific remarks:
L156: which was the age of the queens?
L177: is it the colour the only parameter used to indicate the presence of phacelia pollen? Any palynological analysis? What about other kind of pollens collected by honeybees?
L179: which kind of observation you done to acquire information about the worker bees’ behaviour in the field (visual, video-camera….)? What about other pollinators observed on phacelia flowers during observation time?
L188-190: please add the producer details of the instruments used. Moreover, why you measured such environmental conditions if any data is provided in the results and/or other manuscript section?
L191-197: this section should be strongly improved as concerns the statistical approach you used. A dedicated paragraph will be welcomed. For example: have you used a General Linear Mixed Model (GLMM)? Could you provide details of the general structure of the model you chosen? As it is presented is it not possible to provide an evaluation of the statistic suitability.

Validity of the findings

Introduction: it is fine. However, I feel that literature on the use of phacelia as supplementary floral resource should be more carefully analyzed, presented and discussed, also in comparison to other plants usually employed at the aim. Therefore, authors should strongly expand the sentences comprised between lines 142-148. Nosema is not a virus (line 68), please change.

Results: please merge “Pollen collection experiment” in the subsequent paragraph.
L205: you are writing about a two-tailed t-test: why it is not described in the material and method section? Please check and provide. Table 1: it is not necessary, adequate information could be provided in the text. Graphs: please provide significant letters, where appropriate.

Discussion: it is sufficient. However, it can be noted that almost all of the suggestions provided in this section about the procedures to be included in agri-environment schemes assessments (cfr. lines 265-277) were not evaluated in the presented research (i.e. points 2, 3, and 4). I feel that this represent the main weakness of the paper, which satisfies just the points 1 and (only partially) 2. On this basis, the research should be considered as preliminary and not exhaustive of the hypotheses that are put to the test.

Additional comments

I have a number of substantial criticisms that prevent me to suggest the publication in the present form. I feel the manuscript should be reworked before resubmitting, to increase the overall quality of the presented research. The manuscript lacks in technical quality, in particular statistic and presentation of results should be strongly improved. A year of observation is not sufficient to provide adequate information on the subject. I strongly encourage the authors to replicate for at least another year the observation and to provide additional data about different pollinators foraging on phacelia and other types of pollen collected by honebees in the same experimental conditions.

Reviewer 3 ·

Basic reporting

no comment

Experimental design

no comment

Validity of the findings

no comment

Additional comments

In the manuscript by Sprague et al., is showed how is used the plant P. tanacetifolia in the Agri-Environment Schemes. The work is interesting and well conceived/writed, however some parts are not complete.

The statistical analysis used should be clearly described in the sub-paragraph of M&M section. Also the output of any statistical analysis (in terms of statistical value, degree of freedom and p value) should be indicated in the results section and/or graph caption.

In my opinion by changing the manuscript as suggested, it will be free to publish soon.

---

## Round 0.2 · accepted · Accept

The revised manuscript can be accepted now.

Reviewer 2 ·

Basic reporting

No comments

Experimental design

No comments

Validity of the findings

No comments

Additional comments

Dear colleagues,
thanks for accepting almost all of my suggestions. The manuscript appears much improved. However, I feel that the main criticism of the research remains the low amount of data (just one year of observations). The Editor will decide on this point.

Reviewer 3 ·

Basic reporting

no comment

Experimental design

no comment

Validity of the findings

no comment

Additional comments

Just a little correction: in 215 there are a double dot that could be deleted.
Based on the new version the manuscript can be considerate as ready for publish.